# The Molecular and Pathophysiological Functions of Members of the LNX/PDZRN E3 Ubiquitin Ligase Family

**DOI:** 10.3390/molecules25245938

**Published:** 2020-12-15

**Authors:** Jeongkwan Hong, Minho Won, Hyunju Ro

**Affiliations:** 1Department of Biological Sciences, College of Bioscience and Biotechnology, Chungnam National University, Daejeon 305-764, Korea; ghdwjdrhks1@naver.com; 2Biotechnology Process Engineering Center, Korea Research Institute of Bioscience & Biotechnology (KRIBB), 30 Yeongudanji-ro, Cheongwon-gu, Cheongju 28116, Korea

**Keywords:** LNX, PDZRN, RING, PDZ domain, E3 ubiquitin ligase, UPS, lysosomal pathway, diseasefluorescent probe, Nile-red, biothiols, living cells, theoretical calculations

## Abstract

The ligand of Numb protein-X (LNX) family, also known as the PDZRN family, is composed of four discrete RING-type E3 ubiquitin ligases (LNX1, LNX2, LNX3, and LNX4), and LNX5 which may not act as an E3 ubiquitin ligase owing to the lack of the RING domain. As the name implies, LNX1 and LNX2 were initially studied for exerting E3 ubiquitin ligase activity on their substrate Numb protein, whose stability was negatively regulated by LNX1 and LNX2 via the ubiquitin-proteasome pathway. LNX proteins may have versatile molecular, cellular, and developmental functions, considering the fact that besides these proteins, none of the E3 ubiquitin ligases have multiple PDZ (PSD95, DLGA, ZO-1) domains, which are regarded as important protein-interacting modules. Thus far, various proteins have been isolated as LNX-interacting proteins. Evidence from studies performed over the last two decades have suggested that members of the LNX family play various pathophysiological roles primarily by modulating the function of substrate proteins involved in several different intracellular or intercellular signaling cascades. As the binding partners of RING-type E3s, a large number of substrates of LNX proteins undergo degradation through ubiquitin-proteasome system (UPS) dependent or lysosomal pathways, potentially altering key signaling pathways. In this review, we highlight recent and relevant findings on the molecular and cellular functions of the members of the LNX family and discuss the role of the erroneous regulation of these proteins in disease progression.

## 1. Introduction

Protein ubiquitylation, which is a highly conserved post-translational modification in which ubiquitin (consisting of 76 amino acids) is covalently linked to lysine residues of a substrate protein, is associated with nearly all aspects of eukaryotic physiology. The number of genes encoding proteins involved in ubiquitin-dependent post-translational modification in humans is estimated to be over 1000, which far exceed the total number of kinase-encoding genes (518) identified thus far [1,2]. Protein ubiquitylation comprises a series of enzymatic reactions that initially uses energy from ATP hydrolysis and is mediated by ubiquitin-activating enzyme (E1), ubiquitin-conjugating enzyme (E2), and ubiquitin ligase (E3), which ultimately render substrate specificity [3,4]. E3 ubiquitin ligases can be classified into three groups based on the presence of discrete domains. While HECT family of E3 ubiquitin ligases acts as a genuine enzyme by exploiting its catalytic cysteine residue located in C-terminal HECT domain, the E3 ligase with RING or U-box domain serves as a scaffold that recruits the E2 enzyme in close proximity to the bound substrate. While protein turnover mediated by the UPS is essential for cellular proteostasis, nonproteolytic ubiquitin modification is important for the regulation of epigenetic control, transcriptional regulation, cellular trafficking, and post-translational modification [5]. The presence of extracellular 20S proteasome in blood plasma and cancer cell culture conditioned media indicates the complexity of proteostasis [6]. Therefore, it is not surprising that UPS dysfunction is associated with multiple human diseases. Thus far enormous efforts have been dedicated to understanding the pathophysiological roles of individual E3s because the complexity of the UPS mostly stems from multiple substrates or diverse cellular roles of a single substrate of an E3 ubiquitin ligase [7,8].

Here we describe one of the multifunctional E3 ubiquitin ligases family, ligand of numb protein-X (LNX/PDZRN). Among the five members of the LNX family (LNX1-LNX5), LNX1-LNX4 may function as E3 ubiquitin ligases that target miscellaneous substrates for ubiquitylation, as these possess a RING domain with multiple PDZ (PSD95, DLGA, ZO-1) domains, each of which are considered protein-protein interacting modules [9,10,11,12,13]. Although over 600 E3 ubiquitin ligases are encoded in the human genome [1], LNX proteins belong to the sole E3 ligase family with multiple PDZ domains (up to four); therefore, it is not surprising that members of the LNX family appear to interact promiscuously with various different proteins [14,15,16,17]. Among the LNX proteins, two were originally named LNX1 (also called PDZRN2) and LNX2 (PDZRN1) owing to their ability to bind to NUMB, an intracellular Notch inhibitor [18] via relatively well-conserved NUMB-binding motifs (NPAY in LNX1, NPAF in LNX2) located between the RING domain and the first PDZ domain; the consensus sequences from these are not present in other members of the LNX family. Thus LNX3, LNX4, and LNX5 are more prone to be referred to as PDZRN3, PDZRN4, and PDZRN5 (or PDZRN4L), respectively though (PDZRN; PDZ and RING) [9,17,19,20,21], we will collectively call the several synonyms of LNX/PDZRN/SEMCAP as LNX hereafter.

Although considerable efforts have been devoted during the last two decades to elucidating the function of LNXs, our knowledge of their substrates, binding partners, pathophysiological functions, and detailed molecular mechanisms of action of LNX proteins remains elusive. As an indication of the increased interest in this area of research, an excellent review article published in 2018 [17] comprehensively discussed the role of LNX1 and LNX2 in various diseases. Therefore, in this review, after primarily emphasizing novel discoveries on LNX1 and LNX2 in the last three years, we will focus on findings of the cellular and pathogenic roles of LNX3 and LNX4, and then discuss what major questions remain to be tackled. For the convenience, all the acronyms and their explanations are listed in Table 1.

## 2. Structural Significance and Sequence Similarity of Members of the LNX/PDZRN E3 Ubiquitin Ligase Family

Although the complete X-ray crystallographic structure of LNX proteins is not available, fragmented structures of LNX1p80 and LNX2 are published on various web sites and in the literature. According to previous reports based on the available X-ray crystallographic structures, the N-terminal three-dimensional structures of LNX1 and LNX2 show the presence of a single RING domain with two Zn finger motifs flanking the ends of the RING domain (Zn-RING-Zn fragment), and LNX2 s PDZ domain [22,23,24]. Structural information on the other parts of individual PDZ domains of LNX1 (PDZ2 and PDZ3) are also available in the Protein Data Bank (PDB; http://www.rcsb.org). In 2018 Young [17] provided a detailed account of the structural information of LNX1 and LNX2 that is available. The predicted amino acid sequences and primary structures of human LNX proteins, with domain annotations, are shown in Figure 1. RING-type E3 ubiquitin ligases tend to form homodimers or heterodimers to acquire enzymatic properties, though some act as monomers [25]. Although both LNX1 and LNX2 Zn-RING-Zn fragments could be homodimerized, homodimerization is exclusively required for LNX1 to function as an E3 ubiquitin ligase, while the monomeric status of LNX2 is sufficient for the retention of its auto-ubiquitylation activity [22,24].

The possible formation of an oligomer or heteromer between LNX1 and LNX2 may be considered if we take into account the existence of multiple PDZ domains and class I PDZ-binding motifs (S/T-X-V/I/L) at the C-termini of both proteins (Figure 1a,b). The oligo- or heterodimeric complex formation was validated in yeast two-hybrid experiments, where the presence of C-terminal class I PDZ-binding motifs were a prerequisite for studying their intermolecular interaction [9]. However, whether the Zn-RING-Zn domains of both LNX1 and LNX2 can mediate heterodimerization has not been investigated thus far. Collectively, the dimerization sites in LNX1 and LNX2 may exist in at least two different positions: in the N-terminus containing the RING and Zn finger motifs, and in the PDZ domains. The proposed concepts were strongly supported by the finding that while the introduction of multiple mutations in the gene encoding the isolated Zn-RING-Zn domain was adequate to disrupt homodimerization of the fragments, when introduced in the genes encoding full-length LNX1 and LNX2 the same mutations failed to prevent dimerization [22,24]. Additionally, the intramolecular loop of the C-terminal class I PDZ-binding motifs could also have bound to the second PDZ domains (class I PDZ domain is able to bind S/T-X-C) of LNX1 and LNX2 [15]. Therefore, their intrinsic E3 ubiquitin ligase functions were regulated at the intramolecular level, and could be further modulated by post-translational modifications, such as by the attachment of phosphate moieties. Thus far, only a single protein kinase (c-Src) has been reported to mediate LNX1p80 phosphorylation [26]. In addition, since LNX family besides LNX5 has several Zn ion coordination sites, it would be encouraged to establish the optimal level of Zn ion to achieve as fully active recombinant proteins as possible in vitro by following well established experimental procedures [27,28].

Besides LNX5, the remaining members of the LNX family (PDZRN3 and PDZRN4) contain an N-terminal RING domain and two tandem repeat PDZ domains (Figure 1c–e). While PDZRN3 contains a Zn finger motif at the C-terminal end of its RING domain [29], the presence of this motif in PDZRN4 could not be confirmed, although PDZRN4 does contain a SINA-like domain that may incorporate a Zn ion. Interestingly, the sequences of all members of the LNX family (except LNX1 and LNX2), including their variants, end with a TTV segment, which is also a type of class I PDZ-binding motif. Therefore, LNX1 and LNX2 might interact with other LNX proteins via their PDZ domains and C-terminal class I PDZ-binding motifs. Alternative assumptions might be also possible, considering the structural homology could be inferred from the level of amino acid sequence similarity. Strikingly by pairwise comparison of the individual members of LNX/PDZRN family (Figure 2), we only observed the high sequence similarities between LNX1 and LNX2, and LNX3 and LNX4. While LNX1 and LNX2 as their names imply share some substrates (Table 2) including Numb for ubiquitin dependent modification, LNX3 (PDZRN3) and LNX4 (PDZRN4) though their overlapping targets has not been reported yet would possibly be complementary to each other at least in part under the particular cellular contexts by sharing common substrates. To clarify the unique or overlapping molecular and cellular functions of the individual LNX proteins, their mutually complementary roles on possible target proteins must be scrutinized through future studies.

## 3. Molecular and Cellular Functions of LNX1 and LNX2

Among various binding partners, only specific cytosolic substrates have been reported to be degraded by the UPS after ubiquitylation by LNX1/2, and a relatively large number of membrane proteins appear to be targeted for endocytosis after undergoing direct or indirect LNX1/2-dependent post-translational modifications. Considering the fact that a significant proportion of transmembrane proteins contains the putative PDZ-binding motif at their C-termini, and LNX1 and LNX2 contain four consecutive PDZ domains, it is not surprising that transmembrane proteins appear to be more susceptible to targeting by LNX1/2 than cytoplasmic ones. The newly identified binding targets of LNX1 and LNX2 in the last three years (2018~2020) were listed in Table 2.

One of the most extensively studied substrates of LNX1/2 is Numb, an intracellular Notch signaling inhibitor that can be polyubiquitylated by LNX1/2 for degradation via the UPS [9,19,20,21,22]. Bozozok, also a substrate of LNX2b, regulates the earliest stages of embryonic body patterning [30]. Most recently, Fu et al. [31] reported in 2020 that NEK6 kinase could be subjected to K48-linked polyubiquitylation by LNX1, and subsequently be cleaved by UPS. c-Src, a non-receptor tyrosine kinase, was reported to be a novel degradable substrate of LNX1 that could phosphorylate LNX1 in a transient transfection assay; however, their interrelationship could not be validated in an in vitro assay [26]. A recent report showed an interesting mechanism by which LNX1 indirectly degrades phospho-JAK2 (pJAK2) through a detour mediated by leucine zipper downregulated in cancer 1 (LDOC1) which acts as an adaptor between the two molecules [32].

Members of the melanoma-associated antigen (MAGE) protein family in humans are encoded by approximately 60 genes that are classified into two groups based on their chromosome location and expression pattern [33]. The growing interest in the role of MAGEs in cancer biology has been driven by findings of their elevated expression in cancer tissues [33,34]. Multiple studied have demonstrated that MAGE-assembled protein complexes contain various combinations of MAGEs with RING E3 ubiquitin ligases [33,35]. Accordingly, the complexes are generally referred to as MAGE-RING ligases. Initially, the molecular function of MAGEs was emphasized by their unique roles in augmenting the E3 ubiquitin ligase activity of their cognate RING-containing proteins [33,35,36]. Interestingly, among the constituents of the MAGE family, a type I MAGE-B18 was reported to bind to the LNX1 subdomain localized between the NPAY Numb-binding motif and the first PDZ domain [35]. Thus, it should be analyzed in the future studies whether the MAGE-B18 is able to modulate the E3 ubiquitin ligase activity of LNX1.

Although a recent report showed that LNX1 is primarily localized to the nucleus along with p53 [37], the cellular localization of LNX1 and LNX2 could be altered by the subcellular localization of their binding partners. The endocytosis of LNX1-dependent membrane proteins was initially reported to occur via the redistribution of junctional adhesion molecule 4 (JAM4) from tight junctions to intracellular spaces and its subsequent colocalization with the early endosome marker EEA1 upon stimulation by TGF-β [38]. JAM4 endocytosis was induced by LNX1, and its subcellular redistribution was further facilitated by the presence of Numb. In addition, the TGF-β-triggered endocytosis of JAM4 came to a halt in cells depleted of endogenous LNX1 [38]. However, LNX1-mediated JAM4 endocytosis may not result from the direct ubiquitylation of JAM4 by LNX1, as indicated by the fact that the study was performed using an LNX1p70 isoform, which does not contain an N-terminal RING domain, and therefore, does not exhibit E3 ubiquitin ligase activity [38]. The first LNX1 substrate undergoing endocytosis after ubiquitylation was identified by Takahashi et al. in 2008 [39]. In mammals, the claudin family is composed of 27 members, which have four transmembrane domains and are major constituents of tight junction strands with paracellular barriers and channel functions [40]. The ectopic expression of LNX1 (LNX1p80 with an intact RING domain) was observed to degrade the structures and disrupt the paracellular barrier-related properties of tight junction strands, as confirmed via electron microscopy, by reducing the levels of at least three different claudins (Claudin-1, caludin-2, and claudin-4) at the cell-to-cell junction [39]. LNX1 directly binds to claudins and induces polyubiquitylation, which triggers their endocytosis and subsequent lysosomal degradation.

Coxsackievirus and adenovirus receptor (CAR), which is also an integral membrane protein localized to tight junctions, was reported to interact with LNX1 and LNX2. However, it has not been clarified if the subcellular localization and physiological functions of this protein could be altered by its interaction with members of the LNX family [41,42,43].

Gap junctions are dense arrays of a large number of intercellular channels spanning between the membranes of apposed cells that connect the cytoplasm from the cells, thereby allowing the passage of various small molecules (including ions) as well as electrical pulses. Therefore, junctions are also referred to as electrical synapses [44,45,46]. As the primary constituents of gap junctions, connexins are assembled into connexons, and two connexons are inserted into cell membranes in a head-to-head configuration to form gap junctions. Among the 21 connexins present in humans, connexin36 is most widely expressed in the mammalian brain, similar to LNX1 and LNX2, and participates in the formation of electrical synapses in the central nervous system (CNS) [46,47]. Before 2018, among the 21 connexins identified in humans and 20 in mice, only connexin43 had been shown to be subjected to poly- or multiple monoubiquitylation reactions by various E3 ubiquitin ligases before protein turnover via the UPS-dependent or lysosomal pathways [48,49]. However, it was later shown that the second PDZ domain of LNX1/2 could bind to connexin36 via its C-terminal SAYV motif, presumably at neuronal gap junctions present in the neural tissues of rodents [50]. Both LNX1 and LNX2 induced connexin36 ubiquitylation, and the modified connexin36 did not remain localized to the point of cell-to-cell contact where the gap junctions were formed [50]. After ubiquitylation by LNX2, connexin36 underwent lysosomal-dependent degradation. Therefore, its degradation could be almost completely blocked by treatment with lysosomal inhibitors such as chloroquine or ammonium chloride [50]. The possible interrelationship among LNX1, LNX2, and connexin36-dependent electrical synapse formation in the CNS is characterized by the fact that LNX1 isoforms (LNXp70 and LNXp62), the sole transcriptional variants of LNX1 present in brain tissues, may interfere with LNX2-mediated connexin36 ubiquitylation for the maintenance of neural gap junction integrity, while in other organs, such as the heart, gap junctions composed of connexin36 can be negatively modulated by LNX1p80, the major LNX1 splicing variant present in these tissues, to restrict the levels of electrical synapses generated.

Collectively, LNX1 and LNX2 may play pivotal roles in the remodeling of cell-to-cell junctions by modulating the internalization of various plasma membrane proteins required for the formation of tight junction strands as well as gap junctions. Ubiquitylation-dependent endocytosis for the modulation of cell-to-cell junctional integrity appears to be a general cellular phenomenon, considering that several other cell adhesion-related proteins undergo ubiquitylation before internalization. For example, E-cadherin, which is involved in the formation of adherens junctions, is ubiquitylated by Hakai, a RING-type E3 ubiquitin ligase [51]. Occulin, another constituent of tight junction strands, is ubiquitylated by the HECT-type E3 ubiquitin ligase Itch [52]. Alpha5 integrin can be ubiquitylated and degraded by c-Cbl, a RING-type E3 ubiquitin ligase [53]. The faulty regulation of intramembranous structures may affect cell polarity, cytoskeleton remodeling, intramolecular transport, cell-cell communication, and cellular mobility, which are tightly linked to the acquisition of migratory ability by primary tumors for migration to secondary tissues [54]. Therefore, elaborate processes for the maneuvering of cell junctions is critical for maintaining epithelial tissue integrity and regulating signaling between cells.

Other membrane proteins, such as CD8α, presynaptic glycine transporter (GlyT2), and ephrin type-B receptor 2 (EphB2), which participate in diverse intracellular signaling pathways, could also be targeted by members of the LNX family, and the mechanisms of their regulation by LNX1 and/or LNX2 will be discussed in the following modules in this review [55,56,57,58].

Before ending this module, it should be mentioned that among the various putative LNX1/2-interacting transmembrane proteins, the C-termini of the cytoplasmic domains of certain proteins (Claudin-1, claudin-2, claudin-4, CD8α, and connexin36) contained the di-amino acid sequence YV, which may have contributed protein turnover after ubiquitylation by LNX1 and/or LNX2 [14,15,16,39,50,55]. Though the C-termini contained the same di-amino acid sequence (YV), the longer C-termini sequences can be classified into different classes of PDZ recognition motifs: claudin-1 and CD8α with class III PDZ recognition motif (-X-[D/E/K/R]-X-Φ) contain KDYV and ARYV, respectively; connexin36 and claudin-2 with class II PDZ recognition motif (X-Φ-X-Φ) contain SAYV and TGYV, respectively; and claudin-4 contains SNYV, which cannot be categorized into any reported PDZ recognition motifs (“Φ” represents a single hydrophobic amino acid residue). Therefore, it would be interesting to evaluate whether other putative targets with a C-terminal YV motif could undergo targeted degradation by LNX1/2, which would help determine if this is a generic phenomenon. A likely target with a C-terminal YV motif is the 5-hydroxytryptamine 2B receptor (containing VSYV; Swiss-Prot ID: P41595), which was identified to be an LNX1 binding partner [15].

### 3.1. Recent Findings on the Roles of LNX1 and LNX2 in Synaptic Integrity

The neuronal function of LNX1/2 in adults is primarily related to their roles in maintaining the integrity of both presynapses and postsynapses. Glycinergic interneurons containing the neuronal glycine transporter GlyT2 on their cell surface play a fundamental role in glycinergic neurotransmission, in which glycine, an inhibitory neurotransmitter, is recaptured in the presynaptic terminal [59]. Malfunction of glycinergic neurotransmission caused by loss-of-function mutations in the GlyT2 gene induces neonatal hypertonia and elevated startle responses, followed by muscle stiffness, the symptoms of which are characteristic pathological phenomena in hyperekplexia [59]. The evolutionally well-conserved C-terminal motif of GlyT2 that contains the sequence corresponding to the S/T-X-C PDZ recognition motif (TQC) binds to the second PDZ domain of LNX1/2 [15,58]. GlyT2 was observed to be highly polyubiquitylated and degraded when LNX1 or LNX2 was expressed concomitantly, and consequently, the glycine reuptake capacity of cells expressing GlyT2 was observed to be significantly compromised. *LNX2* knockdown in primary neurons increased the endogenous levels of GlyT2 protein, which indicates the physiological relevance of the presynaptic functions of LNX proteins [58]. Rocha-Muñoz et al. in 2019 [58] further investigated the mechanism underlying protein kinase C (PKC)-dependent GlyT2 degradation in glycinergic neurons, the endogenous expression levels of which are primarily regulated by LNX2, as indicated by the finding that cells depleted of LNX2 failed to respond to treatment with phorbol ester, a well-known PKC activator that has been routinely used to reduce the levels of presynaptic GlyT2 [60,61].

Although LNX1 was barely detected in the level of the total brain, its expression from postnatal week 1 to adult stages in mice remained restricted to hippocampal CA3 pyramidal neurons [56]. In terms of cellular localization, LNX1 is detected exclusively in postsynaptic hippocampal CA3 neurons, which are innervated by the axons of dentate mossy fibers (MFs) [56]. Since the knockout of both LNX1p80 and LNX1p70 in mice led to several morphological defects in the presynaptic terminal of MF axons projected to the hippocampal CA3 region, functional LNX1 in postsynaptic CA3 neurons appears to play pivotal roles in inducing the connection of functional synapses between MF and CA3 in a retrograde and non-cell autonomous manner. LNX1 interacts with EphB1 or EphB2, which are members of a membrane tyrosine receptor kinase family that are specifically expressed in CA3 neurons, whereas their membrane-bound ligand ephrin-B3 is expressed in MFs [62]. Their interaction in postsynaptic membrane compartments is mediated by the C-terminal PDZ-binding motifs of EphB receptors [56]. Contrary to expectation, while the levels of EphB1 and EphB2 proteins were significantly reduced in the CA3 neurons of *LNX1^-/-^* mice, their stability was restored to normal levels when hippocampal neurons were treated with the proteasome inhibitor MG132 [56]. LNX1p70, a major LNX1 variant present in the brain, may protect EphB proteins from degradation by interfering with the functions of other E3 ubiquitin ligases. However, in peripheral tissues, where LNX1p80 is the predominant variant present, EphB proteins could be destabilized by LNX1; this was confirmed in cell culture studies, wherein EphB2 expression was observed to be significantly reduced in response to the co-expression of LNX1p80 [56]. More importantly, the presynaptic defects observed in the MF axons of *Lnx1*^-/-^ mice were phenocopied in *EphB1*^-/-^ and *EphB2*^-/-^ mice, whereas the MF defects observed in LNX1-depleted mice were reversed when the constitutively active form of *EphB2^F620D/F620D^* was expressed [56]. Therefore, EphB-dependent postsynaptic intracellular signaling is required for proper MF-CA3 synapse formation, and LNX1p70 is necessary to maintain the levels of EphB proteins expressed on postsynaptic membranes.

Liu et al. in 2019 [57] confirmed the pivotal roles played by LNX1p70 in the development of social recognition memory by integrating environmental stimuli in the hippocampus. *Lnx1*^-/-^ mice exhibited impaired social recognition memory and autism-associated behavioral patterns. As the frequency of excitatory postsynaptic current in CA3 neurons was reduced in the *Lnx1* knockout mice, the defects in initial social memory consolidation may have been caused by the impairment of functional synapse formation between the MF and CA3 areas [57]. Therefore, MF-CA3 synapse formation mediated by LNX1 is a critical event for the generation of social recognition memory. At the molecular level, LNX1 acts as a scaffold for the formation of a multiprotein complex by recruiting the two glutamate receptor NMDAR subunits GluN1 and GluN2B along with EphB2 using different PDZ domains of LNX1 [57]. Depletion of LNX1 reduced the levels of both EphB2 and GluN2B without affecting GluN1 stability in CA3 neurons. The findings collectively indicated a novel neural function of LNX1 in the consolidation of initial social memory in the hippocampus, which is mediated by the stabilization of a protein complex comprising EphB2 and NMDAR, the core components of memory formation, in the CA3 postsynaptic membrane by LNX1 [63,64].

### 3.2. Role of LNX1 and LNX2 in Tissue and Organ Specification, and Vertebrate Development

The functions of zebrafish Lnx2a and Lnx2b in early embryos are important for the formation of the three perpendicular body axes (dorso-ventral, anterio-posterior, and left-right axes) as well as pancreas organogenesis. We previously showed that the gene encoding Bozozok (a homeobox-containing transcriptional repressor), which is the earliest responsive gene in maternal Wnt stimuli, undergoes UPS-dependent protein degradation after polyubiquitylation by Lnx2b E3 ubiquitin ligase [30,65,66,67]. By destabilizing Bozozok, Lnx2b delimits the boundary of dorsal organizer formation to ensure the proper progression of dorso-ventral axis formation [30,67]. We also showed that Lnx2b, when coexpressed with Cdx4 after the onset of gastrulation, represses *cdx4* transcription by stabilizing the transcriptional repressor complex consisting of Groucho (Gro)/TLE, Hdac1, and Tcf3, irrespective of its E3 ubiquitin ligase activity [68]. E4f1, a multifunctional transcriptional factor, also known as an atypical E3 ubiquitin ligase, upregulated *cdx4* expression by destabilizing the formation of the transcriptional repressor complex by hindering the association between Lnx2b and Hdac1 [68,69]. As indicated by the relationship between E4f1 and Lnx2b in the formation of transcriptional repressor complexes, the posterior body patterning of teleosts may be fine-tuned for caudal development through the tight regulation of *cdx4* expression. CDX4, a homeobox-containing transcription factor, regulates posterior tissue development by inducing the expression of multiple *HOX* genes and promoting erythropoiesis in vertebrates [70,71,72]. The Kupffer’s vesicle (KV), which is formed by a small number of dorsal forerunner cells (DFCs), are small and ciliated organs present transiently during teleost development. The KV is required for the establishment of left-right asymmetry, including laterality in the brain, heart, and visceral organs [73,74,75]. Lnx2b is specifically expressed in DFCs as well as in the KV during zebrafish development. The targeted depletion of Lnx2b in the transient tissues using antisense morpholino delivered into the yolk at the 256–1K cell stages for preventing early dorso-ventral patterning defects led to the randomization of heart jogging and looping, which may be attributed to earlier patterning defects in *southpaw* (a nodal-related gene) in the left lateral plate mesoderm [76,77].

Besides their involvement in early embryonic axis formation, both Lnx2a and its paralog Lnx2b play pivotal roles in the early specification of exocrine pancreas derived from the ventral-anterior bud without affecting endocrine tissues originating from the dorsal-posterior bud, at least in zebrafish [21]. Both Lnx2 paralogs showed distinct expression patterns in the ventral pancreatic bud and redundant functioning in the differentiation of exocrine pancreas by upregulating Notch signaling through the destabilization of Numb and Numb-like proteins. Notably, even though Numb and Numb-like gene transcripts were detected at high levels in the whole pancreas, the proteins were restricted to the dorsal bud-derived primary islet, where Lnx2a and Lnx2b were absent. Therefore, Notch signaling in the ventral pancreatic bud was relieved by the action of Numb and Numb-like proteins [21]. The findings of this study were also valuable because the results of morpholino-based gene knockdown experiments, which are not replicated frequently in null mutants, are simply regarded as off-target effects; however, this method may be accurate in cases wherein the expression of the dominant negative version of an endogenous gene product is necessary [21].

In zebrafish, Lnx2 paralogs may perform versatile functions during early developmental processes; however, Lnx1 does not appear to play any significant role in development, as indicated by its low expression levels during the zygotic stage [30]. However, it is worth noting that the functional analysis of Lnx1 in neuronal development has only been reported in a single study [78]. The low levels of *lnx1* expression may be primarily attributed to the suppressive effects of glycine signaling in zebrafish [78,79]. Defective glycine signaling led to the elevation of Lnx1 levels in the CNS, which in turn prevented the suppression of Notch activity by Numb and promoted neural stem cell proliferation [78].

In contrast to the pivotal roles play by Lnx1/2 in teleost development, the embryonic functions of LNX1/2 in mammals have remained obscure, since LNX1/2 single- or double-knock-out mice did not show any significant embryonic defects and remained fertile, and only exhibited decreased anxiety-related behavior and marginally lower body weight compared to their wild-type counterparts. However, the translated products of LNX1p80 were detected at significant levels in various organs, including the kidney, cecum, colon, and heart, in adult mice, whereas only LNXp70 and LNXp62 were detected in the total homogenates of rodent brain [50,80,81]. Contrary to expectation, the levels of NUMB protein remained unaltered in LNX1/2 double-knockout mice; therefore, the mild defects observed might have been caused by the altered functions of various LNX1/2-interacting proteins besides NUMB [81]. Lenihan et al. in 2017 [81] identified a variety of LNX1/2 binding partners using mass spectrometry, of which several proteins were involved in presynaptic and neuronal functions.

### 3.3. Recent Findings on the Pathogenic Roles of LNX1 and LNX2

The E3 ubiquitin ligase activity of human LNX2, unlike that of LNX1, has been addressed relatively recently, after its ability to bind to the T-cell co-receptor CD8 α-chain (CD8α) via the cytoplasmic C-terminal class III PDZ-binding motif (ARYV, X-D/E/K/R-X-Φ; Φ—hydrophobic residue) of CD8α was confirmed [13,55]. LNX2-dependent CD8α polyubiquitylation is a prerequisite for its degradation via endocytic trafficking to lysosomes. Likewise, LNX1 binds to and attaches multi-monoubiquitin to the cytoplasmic segment of CD8α, which subsequently undergoes lysosome-dependent degradation [55]. These findings are examples of the overlapping functions of LNX proteins that target identical substrates in the same degradation pathway. In addition, it would be interesting to determine whether the dysregulation of LNX1/2 could result in the development of immune diseases owing to the impairment of CD8α localization to the plasma membrane.

In specific studies, LNX1 and LNX2 were shown to be associated with infectious diseases, including Kawasaki disease and Q fever [82,83]. However, the mechanistic explanations for such associations are limited; therefore, the LNX-dependent progression of infectious diseases should be investigated in future studies. A recent finding showed that LNX1 suppressed the aggravation of tuberculosis, an infectious disease caused by *Mycobacterium tuberculosis* [31]. The upregulation of microRNA-325-3p (miR-325-3p), which specifically targets the 3′ UTR of mouse *Lnx1* mRNA, was observed in a mouse model as well as in cell lines, including primary macrophages, infected with *M. tuberculosis*, and may have aided the detour of *M. tuberculosis* to the immune escape pathway [31]. The inverse correlation between miRNA-325-3p and *Lnx1* mRNA expression was striking; additionally, *mir325*-knockout mice exhibited *Lnx1* enrichment and resistance to *M. tuberculosis* infection [31]. NEK6 is a serine/threonine protein kinase that is ubiquitously expressed in various tissues and has STAT3-targeting capability. Following phosphorylation by NEK6, STAT3 prevents the apoptosis of macrophages infected with *M. tuberculosis* by promoting the expression of the anti-apoptotic gene *Bcl2,* which further promotes the intracellular survival of *M. tuberculosis* [31,84,85,86]. NEK6 was observed to be destabilized by LNX1, which directly bound to NEK6 and induced K48-linked polyubiquitylation at K174. Polyubiquitylated NEK6 was subjected to UPS-mediated degradation, owing to which it was unable to phosphorylate STAT3, which is a prerequisite for BCL2 expression [31]. Collectively, these data strongly suggest that miR-325-3p potentiated the intracellular survival of *M. tuberculosis* by negatively regulating the expression of *Lnx1* mRNA at the posttranscriptional level, which resulted in the stabilization of NEK6 and the subsequent activation of STAT3, which is essential for the survival of host macrophages by apoptosis avoidance.

### 3.4. Role of LNX1 and LNX2 in Tumorigenesis

The possible involvement of LNX1/2 in carcinogenesis has garnered interest after the expression levels of LNX1/2 were found to be altered in various tumors. LNX1 assists the tumor-suppressive function of LDOC1 by mediating the targeted degradation of activated JAK2 (pJAK2), and resultantly, interferes in the IL-6/JAK2/STAT3 signaling axis [32]. LDOC1 acts as a bridging molecule between LNX1 and pJAK2 to position them in close proximity for the efficient ubiquitylation of pJAK2 by LNX1 [32]. While the progression of colorectal carcinoma can be regulated by both LNX1 and LNX2, they act in opposing manners. While elevated levels of LNX2 were shown to promote the oncogenic potential of colorectal carcinoma through the maintenance of NOTCH and WNT signaling, both of which are required for cancer cell proliferation [87], LNX1 was observed to negatively regulate cancer stemness in colorectal carcinoma cells, in which LNX1 expression and CAR expression were inversely correlated, and *CXADR* (CAR-encoding gene) expression was shown to be inhibited upon treatment with tamoxifen, a drug frequently used in breast cancer chemotherapy [88] that stimulates LNX1 expression in the presence of estrogen receptor [89]. Contrary to expectation, Camps et al. in 2013 [87] showed that LNX2 depletion reduced the levels of both NUMB and NOTCH1 proteins without affecting the levels of *NUMB* transcripts in SW470 (colorectal cancer cell line) cells, which was counterintuitive, considering that Lnx2 in zebrafish is commonly known to target Numb degradation [21]. Therefore, the oncogenic function of LNX2 mediated via the maintenance of NOTCH signaling in colorectal carcinoma might not be dependent upon NUMB protein, and the degradation of NUMB triggered by LNX2 depletion could be a secondary effect exerted by unknown regulators whose expression or activities are fortified or de-repressed at low levels of LNX2 expression.

The tumorigenic roles of LNX1 have also been observed in other cancer tissues, including gliomas, nervous system tumors, and epithelial-like tumor cells [90,91,92]. Using cDNA microarray analysis, Chen et al. in 2005 [90] found that the levels of *LNX1* transcript reduced significantly in human glioma specimens (collected from 18 patients), based on which they proposed that *LNX1* mRNA levels would serve as a valuable diagnostic marker for glial tumors. Over and above the plausible tumor-suppressive roles of *LNX1* in glioma development [90], various missense mutations in *LNX1* have been frequently observed in neurological tumors in humans [91]. However, it has not been analyzed whether the progression of glial tumors is contingent upon the altered amino acid sequences in *LNX1* (A227V and R428H) [91]. Notably, in 2008 Blom et al. [91] reported that *LNX1* mRNA levels were compromised, and *LNX1* copy numbers at chromosome loci 4q12 were amplified in a significant proportion of tumor specimens, including glioblastoma specimens, which was contradictory to the findings of a previous report [90]. However, considering that gene amplification at the chromosome level does not always correlate with increased transcription, the reduced expression level of *LNX1* in gliomas may have simply resulted from genetic instability at locus 4q12, where *LNX1* is located. The possible correlation between genetic instability at locus 4q12 and *LNX1* expression should be analyzed thoroughly in future studies.

In general, in most solid tumors, the presence of a variable degree of epithelial characteristics correlates inversely with malignancy. Kohn et al. in 2014 [92] observed via meta-analysis that in tumors with epithelial characteristics, LNX1 expression significantly correlated with the expression of selectively expressed oncogene products involved in the formation of tight junctions. Therefore, *LNX1*, in conjunction with other selectively and mutually expressed genes, could be used as a marker for solid tumors with traces of epithelial features [92]. Another recent meta-analysis showed that *LNX1* was hypomethylated, and therefore, was expressed at high levels in patients with nasopharyngeal carcinoma [93]. Therefore, the methylation status of *LNX1* could be used as a valuable diagnostic and prognostic marker for nasopharyngeal tumors.

Although there is an increasing body of evidence on the possible tumor-suppressive roles of LNX1, its novel bona fide oncogenic function was recently demonstrated in a study where LNX1 was shown to indirectly mediate the destabilization of p53, the most well-known tumor suppressor [37,94]. Although several tumors can be developed upon the inactivation of mutations in *p53*, tumorigenesis can also be triggered via the functional suppression of wild-type p53 [94]. For instance, tumors with elevated levels of MDM2, an E3 ubiquitin ligase that mediates p53 degradation, showed significantly reduced cellular levels of p53 irrespective of the mutational status of the gene [94,95,96]. NUMB, which is a functional substrate of LNX1, prevents the ubiquitylation of p53; therefore, the silencing of NUMB accelerates p53 degradation [97]. NUMB-dependent p53 stabilization may be attributed to the ability of NUMB to form a binary complex with MDM2 or p53, which compromises the interaction between p53 and MDM2 [97,98,99]. A possible relationship between LNX1 and p53 was recently proposed after a meta-analysis revealed that in several different cancers, LNX1 expression levels are often higher in tumors with wild-type p53 than in those with mutant p53 [37]. LNX1 was observed to interact with and increase the levels of p53 ubiquitylation [37]. In addition, LNX1 was also observed to bind to MDM2 [37]. After the formation of the LNX1-p53 protein complex, LNX1 did not directly induce p53 ubiquitylation; instead, LNX1-dependent p53 destabilization was indirectly mediated by the degradation of NUMB, which interfered with the MDM2-p53 interaction. The oncogenic potential of LNX1 was further confirmed by the high number of cells with wild-type p53 in the sub-G1 population among *LNX1*-knockout A549 cells, which indicated accelerated apoptosis in these cells [37]. The oncogenic role of LNX1 was also confirmed in xenograft studies, where *LNX1*-knockout A549 cells failed to form any observable tumors in nude mice [37]. Therefore, the data strongly suggest the potential therapeutic value of LNX1, at least in tumors retaining functional p53, based on which LNX1-targetable drugs could be developed for treating patients with cancer retaining wild-type p53.

Collectively, the contradictory cellular effects of LNX1 on tumor growth as an oncogenic or tumor suppressor protein may reflect its distinct functions in discrete cellular contexts, where the mutational status of p53 would influence the role of LNX1 as an oncogenic or tumor-suppressive protein. Therefore, it would be necessary to scrutinize both the expression and the mutation states of p53 in tumor tissues where LNX1 appears to act as a tumor suppressor. 

## 4. Molecular Functions of LNX3/PDZRN3/SEMCAP3

Among the members of the LNX/PDZRN family, LNX3 (also known as PDZRN3 or SEMCAP3, hereafter referred to as LNX3) has been highlighted in recent years after the discovery of its cellular and pathophysiological roles in various tissues and in tumor development. LNX3 was initially identified by in silico prediction [11] and was shown to have two PDZ domains immediately following an *N*-terminal RING domain and a Zn finger domain. Information regarding its isoforms is available in the NCBI website (Figure 1c). Owing to an alternative splicing event, mouse LNX3 can form at least two splicing variants containing a single PDZ (PDZRN3A) or two successive PDZ domains in the central region (PDZRN3B) [29]. Similar to other LNX proteins (LNX1 and LNX2), LNX3 may bind several proteins through its PDZ domains, which allows the recognition and scaffolding of signaling molecules for tight regulation, and consequently, may act as a post-translational modulator by conjugating ubiquitin moieties with its substrates. Although LNX3 was initially reported to lack E3 ubiquitin ligase activity based on findings from in vitro studies [100], other reports have showed that LNX3 acts as a conventional E3 ubiquitin ligase, with its intrinsic E3 activity dependent upon the presence of a functional RING structure [29,101]. The discrepancies could be attributed to the usage of different fragments of LNX3 in in vitro ubiquitylation assays. While Ko et al. in 2006 [100] used an N-terminal LNX3 fragment containing the RING domain and the Zn finger motif (amino acids 1–256), Lu et al. in 2007 [29] used only the RING domain of LNX3 fused to a GST-Flag tag to measure its E3 ubiquitin ligase activity in an in vitro system. Considering that the E3 ubiquitin ligase activity of LNX1 and LNX2 was shown to be influenced by the closely neighboring sequences of the RING domain, it would be interesting to assess whether the full-length construct of LNX3 recombinant protein exhibits E3 ubiquitin ligase activity. For instance, while both Zn finger motifs located at the border of the *N*- and *C*-termini of the RING finger domain are indispensable for conferring E3 ubiquitin ligase activity to LNX1 [24], in case of LNX2, only the N-terminal Zn finger motif is required [22].

Notably, while LNX1 and LNX2 contain two Zn finger motifs close to the RING finger domain, LNX3 and LNX4 only have one Zn finger motif immediately next to the RING domain [24]; therefore, further investigation is necessary to confirm whether the single Zn finger motifs of LNX3 and LNX4 contribute to the regulation of their E3 ubiquitin ligase activity. In addition, given the structural significance of the LNX/PDZRN family owing to the presence of multiple PDZ domains, C-terminal class I PDZ-binding motif ((D/E)-S/T-X-Φ), and RING domain-mediated dimerization potential in LNX1 and LNX2 [22,24], heterodimerization or oligomerization between members of the LNX family might impart the E3 ubiquitin ligase activity of the counterpart to the complex, similar to the well described MDM2-regulating function of MDMX (MDM2 homolog), which fortifies p53 degradation by enhancing the E3 ubiquitin ligase activity of MDM2 [95,96,102,103].

Although the RING finger domains of both MDM2 and MDMX are critical for their mutual activity and their action on p53, as well as for heterodimerization, MDMX alone, irrespective of its binding affinity to p53, does not support p53 ubiquitylation and degradation owing to the lack of E3 ubiquitin ligase activity per se [95]. In vitro ubiquitylation experiments by Lu et al. in 2007 [29] revealed that monoubiquitylation by LNX3 occurred only when it was present by itself, similar to MDM2-dependent p53 monoubiquitylation, which occurred only when MDMX was absent [96]. However, other authors did not report the auto-ubiquitylation activity of LNX3 [100]. One of the tentative explanations for this may be that the complete potential of the E3 ubiquitin ligase activity of LNX3 could be bestowed by other E3s, such as homologous members of the LNX family, through the formation of an intermolecular complex via direct interaction, similar to the intermolecular regulatory mechanism in the MDM2/MDMX protein complex. The possible inter- or intra-relationship among members of the LNX family could be comprehensively tested in an in vitro assay with proper substrates. The known substrates of LNX3 are summarized in Table 3. 

### 4.1. Cellular and Pathophysiological Functions of LNX3/PDZRN3/SEMCAP3

LNX3 is widely expressed in several tissues, including neural, skeletal, muscular, and vascular tissues, in vertebrates [29,100,101]. During the early embryonic development of zebrafish and *Xenopus laevis*, the spatiotemporal expression profile of LNX3 alters dynamically following the occurrence of developmental processes in neural and non-neural tissues, including the optic stalk, retina, rhombomere, motor neurons, dorsal midline, somites, and pronephros [104,105]. Among the multiple mesodermal tissues with detectable levels of LNX3 transcripts, the proper expression of LNX3 in the developing pronephros appeared to be necessary as early as during the initial steps of nephrogenesis in *Xenopus laevis* [105]. Depletion of LNX3 using an antisense morpholino led to abnormal morphogenesis, particularly in the nephrostome, and coiling of the proximal tubules, while it mildly affected the duct and glomus formation even when it was expressed in all structural constituents of the functional kidney [105]. Besides being involved in nephrogenesis, LNX3 also helps maintain proper functioning in the proximal tubule. The first PDZ (PDZ1) domain of LNX3 binds to the class I PDZ-binding motif of SMCT1 (SLC5A8), which is reported to be a sodium-coupled monocarboxylate (MC) transporter with seven transmembrane domains essential for the reabsorption of MCs into the renal proximal tubule [106]. The augmentation of SMCT1 transporter activity by the binding of PDZK1, a sodium/hydrogen exchange regulatory factor, was inhibited by LNX3, as it competed with PDZK1 for access to the C-terminal PDZ-binding motif of SMCT1 [106]. However, whether the E3 ubiquitin ligase activity itself of LNX3 contributes to the negative regulation of SMCT1/PDZK1 transporter activity is unclear.

A recent report suggested that LNX3 plays a crucial role in triggering the endocytosis of the Frizzled/Disheveled 3 (Dvl3) complex at the plasma membrane by facilitating direct interaction with and mediating K63-linked ubiquitin conjugation of Dvl3 [101]. The binding of Wnt to its receptor Frizzled, which triggers the Dvl3-dependent signaling cascade, primarily controls the Wnt/planar cell polarity (PCP) pathway, and defects in this process have been associated with renal diseases in humans. This provides compelling evidence in favor of the involvement of LNX3 in cystic kidney disease [107,108]. Interestingly, LNX3 is located on the short arm of chromosome 3. In certain cases of renal tumors, an unbalanced translocation t(3:6) was observed at the breakpoint of 3p12.3 between LNX3 and CNTN3 and at 6q24.3, which contains the seventh intron of *STXBP5* [109]. Although the levels of LNX3 transcripts in tumor cells carrying an unbalanced t(3;6) translocation did not change significantly [109], the plausible effect of LNX3 on the development of conventional renal cell carcinomas should be analyzed thoroughly in future. Of note, a recent report suggested the possible involvement of LNX3 in the process of renal Mg^2+^ reabsorption, which involves the reduction of the abundance of claudin-16 (CLDN16) at the tight junction of the thick ascending limb of the Henle’s loop [110]. Direct binding of LNX3 to the dephosphorylated form of the C-terminal class I PDZ-binding motif of CLDN16 might induce the monoubiquitylation and subsequent endocytosis of CLDN16, following which CLDN-16 would be targeted by the late endosome for degradation [110]. Considering the causative role of CLDN16 dysfunction in the progression of hypomagnesemia [111], LNX3 might play a critical role in the maintenance of renal homeostasis by regulating the proper subcellular localization and levels of CLDN16. In addition, an unbiased genetic approach using a genome-wide association study (GWAS) in patients from an ethnic group with end-stage renal disease showed the presence of a specific single nucleotide polymorphism (SNP) in LNX3, which was strongly associated with sudden cardiac death resulting from arrhythmia on dialysis [112]. Therefore, the data collectively suggested that LNX3 performs versatile functions in early renal development and in the maintenance of renal homeostatic function. It should be analyzed in further studies whether LNX3 transcripts, apart from those present in developing pronephros, also play crucial roles in tissue specification and differentiation during early embryonic development as well as in adult tissues and organs, since the diverse cellular functions of LNX3 have only been confirmed in in vitro cell culture studies.

Besides LNX3 knockdown studies performed using *Xenopus laevis*, the targeted knockout of LNX3 during the angiogenic gestational stage from E7.5, during which mice exhibited embryonic lethality, induced massive defects in the extraembryonic vasculature, while conditional deletions in the gene in the endothelium in neonates caused severe blood vessel defects with loss of polarity in endothelial cells, which is dependent upon Wnt/PCP transduction, a process in which LNX3 activates non-canonical Wnt signaling by triggering the endocytosis of Frizzled/Dvl3 (Figure 3) [101].

The ectopic expression of LNX3 in endothelial cells also led to early embryonic lethality along with drastic hemorrhage and malformation of endothelial intercellular junctions [113]. The Par3-Par6-atypical PKC polarity protein complex establishes apico-basal polarity by functionally interacting with a large number of proteins associated with cell polarization, which determines the integrity of the intercellular tight junction and adherence junction along the apico-basal axis in the cellular membrane [114,115]. PKCζ, an atypical PKC localized to the cellular junction of epithelial and endothelial cells, reportedly regulates endothelial cell permeability [116,117,118,119,120]. The potentiation of PCP by the ectopic expression of LNX3 in endothelial cells activates PKCζ and c-jun signaling, which leads to rupturing of the blood-brain barrier (BBB) owing to the destabilization of intercellular endothelial junctions (Figure 3) [113].

At the molecular level, the destabilization of MUPP1 by LNX3 in a UPS-dependent manner exerted a negative effect on the formation of a polarity complex composed of PAR3, PKCζ, and MUPP1 at the intercellular junctions (Figure 3) [113]. Therefore, the downregulation of Wnt/PCP signaling would be advantageous for preventing BBB rupturing, such as during ischemic stroke. Notably, the PDZ domains of LNX1 and LNX2 exhibited significant homology with the four C-terminal PDZ domains of MUPP1, and these were considered to share a common ancestral gene [121]. More importantly, LNX3 appeared to act as a molecular switch between canonical and non-canonical Wnt signaling. Sweduth et al. in 2014 [101] showed that while the Wnt/β-catenin pathway was fortified by the depletion of LNX3, Wnt/PCP signaling was concomitantly inhibited. In addition, the canonical Wnt-inhibitory role of LNX3 was further validated by the finding that LNX3 depletion potentiated BMP-2 dependent osteoblast differentiation of C2C12 mouse mesenchymal progenitor cells through the activation of Wnt/β-catenin signaling [122]. Collectively, these data are reminiscent of the scaffolding function of Lnx2b as a canonical Wnt/β-catenin antagonist, which involves the stabilization of the transcriptional repressor complex composed of Tcf3, Hdac1, and Gro/TLE [68]. Therefore, LNX3, in coordination with LNX proteins, may play versatile roles in various tissues where they are expressed by regulating at least one pivotal cellular signaling pathway, such as Wnt signaling (Figure 3).

The mechanism underlying the physiological action of LNX3 at the cellular level has been investigated extensively for its role as a cell-fate modulator in cellular differentiation at different stages in a cellular context-dependent manner. There are four myogenic regulatory factors (MRFs), Myf5, MyoD, myogenin, and Mrf4 (Myf6), all of which contain a basic helix-loop-helix (bHLH) domain, which imparts transcriptional activity [123]. The stepwise onset of expression of individual MRFs is a critical cellular event for the commitment of myogenic precursor cells as well as the terminal differentiation of myoblasts into skeletal myocytes and their fusion into myotubes [124,125]. LNX3 was initially reported to be a key regulator of terminal myogenic differentiation in C2C12 myoblasts [100,126]. During terminal myogenic differentiation, LNX3 expression was elevated following the induction of myogenin, which governs the late phase of myogenesis yet precedes the expression of myogenin heavy chain (MHC) [100,126,127]. Interference with C2C12 myoblast fusion for the formation of multinucleated myotubes resulting from LNX3 knockdown occurred concurrently with the reduction in MHC expression levels. However, the cellular levels of MyoD and myogenin remained constant after the stimulation of myogenic differentiation regardless of the LNX3 expression levels [100]. LNX3 depletion increased the expression level of Id2, which forms a heterodimer with myogenin and inhibits its transcriptional activity by interfering with its promoter-binding ability [126,128,129]. Therefore, the inhibitory effects of LNX3 depletion on the differentiation of myoblasts into myotubes could be attributable, at least in part, to the interference in myogenin function resulting from Id2 upregulation along with the augmented expression of STAT5b, which appears to bind to and activate the Id2 promoter [126]. This was further validated by the finding that the simultaneous depletion of LNX3 and Id2 led to the restoration of MHC levels [126]. In addition to the imperative role of LNX3 in the late phase of myogenic differentiation, the anti-apoptotic function of LNX3 was also reported to be necessary for maintaining the oscillation of cyclin A2, a ubiquitously expressed A-type cyclin found in proliferating cells, and consequently, for ensuring genomic stability [130,131,132]. In contrast, another report showed that the expression level of p53 reduced significantly in HeLa cells, which constitute a cervical carcinoma cell line with multiple copies of integrated high-risk human papillomavirus-18 (HPV-18) DNA [133], when LNX3 was depleted using an siRNA [134]; therefore, the increased cellular levels of LNX3 might induce cell cycle arrest under different cellular contexts.

The cellular role of LNX3 as a key cell-fate determinant was also indicated by its involvement in the differentiation of mesenchymal progenitor cells into myotubes, as well as into osteoblasts and adipocytes, in contradictory manners. In contrast to the differentiation of myoblasts into myotubes, which requires functional LNX3 [100,126], the inhibition of LNX3 was necessary for both BMP-2-induced osteoblast differentiation of C2C12 cells [122] and adipogenesis of 3T3-L1 preadipocytes through the augmentation of STAT5b expression, through which the transcription of PPARγ, a master regulator of adipogenesis, is stimulated [135,136,137]. Notably, while LNX3 expression levels must be maintained or significantly reduced for the differentiation of mesenchymal stem cells into discrete cell lineages, LNX3 overexpression failed to influence cell-fate determination. Therefore, LNX3 only plays permissive roles, and not instructive roles, in the differentiation of mesenchymal stem cells, which may express LNX3 at levels sufficiently above the threshold level. In contrast, the instructive role of LNX3 was demonstrated in mouse skeletal muscle tissues with underdeveloped neuromuscular junctions by the reduction of muscle-specific receptor tyrosine kinase, a key regulator for all aspects of postsynaptic differentiation, in response to LNX3 overexpression [29,138]. Therefore, the cellular function of LNX3 might be dependent on the level of expression and status in different cellular contexts. Thus far, it is unclear whether the E3 ubiquitin ligase activity of LNX3 is a prerequisite for the determination of mesenchymal stem cell fate.

### 4.2. Other Pathological Roles of LNX3/PDZRN3/SEMCAP3, Particularly in Tumor Development

An SNP of LNX3 in intron 3 associated with grade 4 neutropenia was identified through a GWAS of patients who suffered from severe neutropenia after treatment with irinotecan, a chemotherapeutic agent for the treatment of colorectal and lung cancers [139,140,141,142]. Considering the cell cycle regulatory function of LNX3, it would be reasonable to assume that its altered expression might correlate with cancer development. The possible involvement of LNX3 in neoplasia was postulated based on findings from a GWAS using various cancer specimens. For instance, LNX3 expression levels were significantly reduced in biliary tract cancer [143]. Interestingly, several recent reports have showed that in cancer patients, the chromosomal translocation t(3;3)(p13;p25) resulted in various in-frame gene fusions between LNX3 and RAF-1 proto-oncogene kinase [144,145,146,147,148]. For example, one of the fusion proteins expected to form via the chromosomal rearrangement between exon 5 of LNX3 and exon 8 of RAF1 appeared to possess a functional RING finger motif along with the first PDZ domain, followed by the intact kinase domain of RAF-1 without the *N*-terminal RAS-binding and C1 auto-inhibitory domains [145]. Although the LNX3-RAF1 fusion protein may undergo constitutive activation, following which it can ceaselessly activate downstream MAPK pathways, the detailed molecular mechanisms underlying tumor development should be addressed in future studies.

The E6 proteins of high-risk human papillomaviruses (HPVs) with high oncogenic potential have a C-terminal PDZ-binding motif, deletion of which is associated with viral genome integration and potentiation of oncogenic transformation [149,150,151]. LNX3 was isolated as one of the binding partners of the HPV-16 E6 C-terminal peptide through high-throughput proteomic analysis [152,153]. Using their *C*-terminal PDZ-binding motifs, HPV-16 and HPV-18 E6 directly bind to and induce the proteasome-dependent degradation of LNX3 [134]. It appears to be counterintuitive that an E3 ubiquitin ligase that would be highly unstable owing to its auto-ubiquitylation activity could serve as a target for facilitated degradation by another protein. However, considering that the proteolytic clearance of several tumor suppressor proteins, including p53 and Scribble, via the recruitment E6AP (a HECT domain-containing E3 ubiquitin ligase) is the predominant function of HPV E6 [154,155,156,157,158], it is not surprising that LNX3 could be subjected to targeted degradation by E6AP bridged via the E6 scaffold protein. Since LNX3 downregulation is a prerequisite for the differentiation of mesenchymal stem cells into cells of discrete lineages [122,136,137], it should be assessed if the introduction of E6 could trigger the cellular differentiation of stem cells.

To date, several interacting proteins of LNX3 have been reported in different tissues and cell lines, which might be suggestive of the multifarious cellular functions of LNX3 in distinct signaling cascades. Given that LNX3 or its splicing variant, at least in mice, contains a RING finger, a single Zn finger, one or two PDZ domains, and a class I PDZ-binding motif at the C-terminal end, it may interact with various proteins and regulate their cellular function and fate primarily through the attachment of ubiquitin moieties to their binding partners, or may provide a scaffold for recruiting signaling components in close proximity. It would be interesting to study the pathophysiological relevance of LNX3 in tumor development to determine whether LNX3 could serve as a candidate biomarker or a target for molecular therapies.

## 5. Possible Pathophysiological Roles of LNX4/PDZRN4

Besides LNX5 (also known as LNX4L, PDZRN4L, PDZRN5), whose functional analysis has not been performed thus far, LNX4 (also known as PDZRN4) is the least studied member of the LNX/PDZRN family that was initially discovered through in silico studies [11,12]. LNX4, similar to other proteins of the LNX family (except LNX5, which lacks the RING domain and may not act as an E3 ubiquitin ligase), contains an N-terminal RING that is immediately followed by a putative Zn finger depicted as the SINA homolog motif, two PDZ domains, and a C-terminal class I PDZ-binding motif (Figure 1d). The sequences of various isoforms of LNX4 that lack the RING domain have been aligned in Figure 1d. Although LNX4 is considered an E3 ubiquitin ligase owing to the presence of the RING domain, its ubiquitin ligase activity and binding partners have not been characterized to date. Given the high sequence homology between LNX3 and LNX4 (Figure 2c), it would be worth to test whether they can share substrates, if any, for ubiquitylation in similar or discrete manners.

The possible pathophysiological roles of LNX4 were initially analyzed in hepatocellular carcinoma (HCC), which is the major subtype of liver cancer that accounts for approximately 80% of all cases [159,160]. A significant reduction or complete absence of LNX4 expression was observed in two-thirds of human HCC specimens and in all established HCC cell lines (13 lines) [160]. More importantly, since overexpression of LNX4 inhibited the proliferation of three independent HCC cell lines, LNX4 may act as a tumor suppressor in HCC [160]. The tumor-suppressive function of LNX4 was further corroborated in another recent report, which demonstrated that LNX4 expression was significantly compromised in breast cancer tissues at both mRNA and protein levels [161]. In addition, low LNX4 expression correlated with malignant progression and poor survival rates in patients with breast cancer. While LNX4 knockdown was associated with the facilitation of breast cancer cell growth and metastasis in in vitro and in vivo studies, overexpression of LNX4 was sufficient to suppress cell invasion and proliferation [161]. Therefore, low LNX4 expression levels appear to be closely correlated to aggressive clinicopathological characteristics in breast cancer, and may serve as a valuable prognostic marker for patients with breast cancer. In addition, LNX4 expression also reduced drastically in rectal adenocarcinoma tissues, and its downregulation appeared to correlate with the increased methylation and upregulation of hsa-mir-182 expression, which may target LNX4 transcripts [162]. To determine whether the tumor-suppressive action of LNX4 is a standard phenomenon, such studies should be performed for other cancer types as well.

The close relationship between frequent LNX4 mutations and tumor development was highlighted through whole-exome sequencing and whole-genome sequencing of samples collected from patients with brain metastasis of colorectal cancer [163]. The highly recurrent rate of LNX4 mutations compared to that of mutations in other LNX proteins in the COSMIC database further confirms its functional significance as a tumor suppressor. In contrast, LNX4 expression level was found to be significantly elevated in patients with papillary thyroid carcinoma resulting from heavy radiation exposure during childhood [164]. Although the mechanistic explanation remains vague, the increased expression of LNX4 caused by radiation exposure during childhood may have resulted from feedback responses for the suppression of tumor development, at least in the case of papillary thyroid carcinoma.

A hitherto undiscovered disease-associated role of LNX4, apart from its tumor-suppressive function, was proposed based on the relationship between an SNP in LNX4 and increased susceptibility to multiple sclerosis (MS), a chronic inflammatory demyelinating disorder of the CNS [165,166]. Although it remains controversial, it has been postulated that LNX4 could be used as a novel biallelic MS marker [165]. However, since limited information is available on the functional significance of LNX4 at the molecular level owing to the unavailability of model animals with altered LNX4 expression, it is important to use genetically modified animals to investigate the molecular and physiological roles of LNX4 in early embryonic development and the pathogenic mechanisms adopted by LNX4.

## 6. Conclusions

There are five members of the LNX/PDZRN family, four of which contain a RING finger domain and multiple PDZ domains. They bind to and modify with ubiquitin multitudinous proteins including several key signaling components by direct or indirect manner. The overlapping functions of LNX proteins are yet to be clarified; however, there are evidences that at least a few common substrates can be targeted by different LNX proteins in a complementary manner. For instance while LNX1 and LNX2 with high sequence similarities (Figure 2b) somewhat seem to share the E3 ubiquitin ligase function by targeting same substrates, LNX3 and LNX4 also shows relatively high sequence homology though (Figure 2c), their common substrate proteins have not been identified yet. The unique expression profiles of individual LNX proteins and their varied cellular localization patterns are noteworthy. For instance, LNX1 is primarily localized to the cytosol and to a lesser extent to the nucleus, whereas LNX2 was observed to be an exclusively cytosolic protein, and stabilized LNX3 localized to the nuclei in HeLa cells [55,134]. Therefore, it can be hypothesized that individual LNX proteins might act in mutually supplementary or exclusive manners by sharing substrates in different tissues, cells, and intracellular organelles. It is also noteworthy that while LNX1 and LNX2 are involved in versatile cellular signaling by targeting wide range of proteins, LNX3 seems to primarily govern cell-fate determination through fine-tuning of Wnt/PCP (Figure 3) and BMP signaling in cellular contexts dependent manner. Possible involvement of LNX3 in cancer development was strongly suggested through the observation of cancer patients who have chromosome translocation between LNX3 and RAF-1 proto-oncogene. Though there is limited information on the pathophysiological functions of LNX4, the tumor suppressive role of LNX4 has been gradually unveiled after analyzing various tumor patients and cancer cell lines. Although accumulating evidence has elucidated the pathophysiological relevance of the LNX family, the following questions/aspects must be addressed:(1)What are the molecular and cellular functions of LNX5?(2)What are the substrates of LNX4? Could LNX4 share substrates with LNX3?(3)Which are the post-translational modifiers of LNX proteins that may alter their molecular activities?(4)Why are only selected LNX1/2 substrates targeted by the UPS, whereas several miscellaneous proteins appear to bind to them? Are they all/are most of them LNX-binding proteins that can be modified through ubiquitylation?(5)Complete three-dimensional structures of individual LNX proteins must be validated in future studies for a better understanding of their molecular function.(6)Diverse types of genetically modified animals with altered LNX expression profiles should be established to evaluate the cellular functions of LNX proteins.(7)The developmental significance of LNX1 and LNX2 in teleosts and mammals appears to differ. The source of this difference should be identified.(8)Could LNX proteins serve as potential targets for drugs that interfere with their E3 ubiquitin ligase activity, inhibit their substrate-binding ability, or promote their function to facilitate substrate degradation?(9)Does the loss of N-terminal RING and Zn finger domains of LNX/PDZRN by alternative splicing event affect the function of E3 ubiquitin ligase activities of other variants retaining RING domain? It would be an interesting research topic by considering the discrete expression pattern of LNX1p70 and LNX1p80, and their opposite effects on the stability of EphB2 [56]. In addition, we also previously reported that N-terminal truncation of the endogenous *lnx2a* in zebrafish interfered with not only the E3 ubiquitin ligase activity of Lnx2a but also Lnx2b [21]. Thus it should be scrutinized in the future whether the dominant negative effects of N-terminal truncation of LNX/PDZRN by alternative splicing are critical yet generic mechanisms for fine-tuning of cellular proteostasis to be tightly controlled.

## Figures and Tables

**Figure 1 molecules-25-05938-f001:**
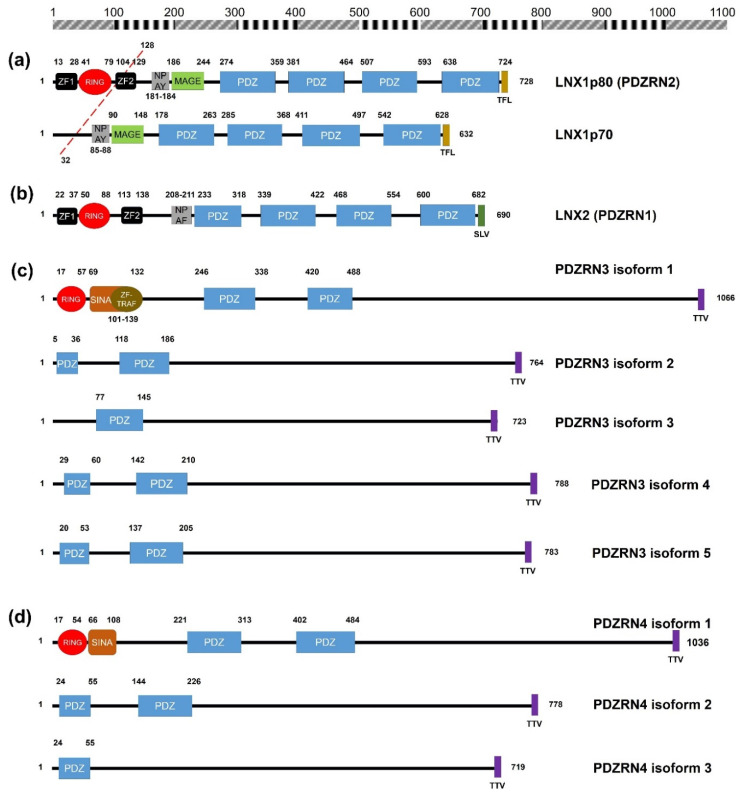
Schematic diagrams of the members of the human LNX family. (**a**) LNX1 has two splicing variants, LNX1p70 and LNX1p80. In contrast to the longer splicing variant, the shorter one does not contain the N-terminal RING domain and the two neighboring Zn finger motifs, while the other known domains following the second Zn finger motif are conserved, which include four PDZ domains and a NUMB-binding motif. Both variants bind to NUMB via the NPAY motif and contain a MAGE-B interacting region. The red dashed bar indicates the point from which the amino acid sequences become identical. (**b**) LNX2 showed the highest similarity with LNX1p80 in terms of primary structure. However, the major difference between LNX1p80 and LNX2 was noted at the region located between the NUMB-binding motif and the point of initiation of the first PDZ domain. While LNX1p80 has a relatively longer sequence that includes the MAGE-B18-interacting motif, LNX2 only has a 21-amino acid-spanning region. (**c**) LNX3 has five splicing variants. The PDZRN4 isoform 1 is the sole variant equipped with an N-terminal RING domain and a juxtaposed Zn finger flanking the C-terminal end of the RING domain. All the variants have two PDZ domains, except the third isoform, which only has one PDZ domain. (**d**) Among LNX4 variants, which include three isoforms, only the PDZRN4 isoform 1 has an N-terminal RING domain and a SINA-like motif. Similar to LNX3, the variants have two PDZ domains; the third isoform is an exception as it has only one relatively short PDZ domain identical to that in the PDZRN4 isoform 3. (**e**) LNX5 has six splicing variants. All variants have a single PDZ without any recognizable domains. The last three single letter amino acid codes represent the class I PDZ-binding motif (S/T-X-V/I/L). The scale bar on the top indicates the approximate length in terms of the number of amino acids for the comparison of individual proteins. ZF; Zn finger motif, RING; really interesting new gene, MAGE; melanoma-associated antigen, PDZ; PSD95-DLGA-ZO-1, SINA; seven in absentia, TRAF; tumor necrosis factor receptor (TNF-R)-associated factor. The schematic illustration of the alignment of the LNX/PDZRN domains are derived from the deduced amino acid sequence information retrieved from GenBank; the corresponding accession numbers have been provided: LNX1p80 (NP_001119800.1), LNX1p70 (NP_116011.2), LNX2 (NP_699202.1), PDZRN3 isoform 1 (NP_055824.1), PDZRN3 isoform 2 (NP_001290068.1), PDZRN3 isoform 3 (NP_001290069.1), PDZRN3 isoform 4 (NP_001290070.1), PDZRN3 isoform 5 (NP_001290071.1), PDZRN4 isoform 1 (NP_001158067.1), PDZRN4 isoform 2 (NP_037509.3), PDZRN4 isoform 3 (EAW57827.1), LNX5 isoform 1 (NP_001290441.1), LNX5 isoform 2 (NP_115901.2), LNX5 isoform 3 (NP_001290442.1), LNX5 isoform 4 (NP_001290443.1), LNX5 isoform 5 (NP_001290444.1), LNX5 isoform 6 (NP_001290445.1).

**Figure 2 molecules-25-05938-f002:**
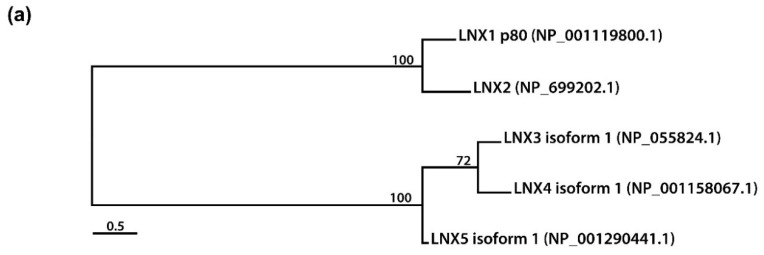
Amino acid sequence comparison of human LNXs (**a**) Phylogenetic tree of the LNX proteins. Maximum likelihood analysis was performed using to RAxML under the LG+GAMMA model. Analysis was carried out with five LNXs encoding the longest of their own variants (LNX1p80, LNX2, LNX3 isoform 1, LNX4 isoform 1, and LNX5 isoform 1). The numbers on each node represent bootstrap supported values. The scale bar indicates the number of substitutions/site. (**b**,**c**) Homology comparison of each domains was highlighted by the value of percentage. (**b**) Pairwise comparison between LNX1 and LNX2. The scale bar on the top indicates the approximate numbers of amino acids for the comparison of individual proteins. (**c**) Pairwise comparison between LNX3 (PDZRN3) and LNX4 (PDZRN4).

**Figure 3 molecules-25-05938-f003:**
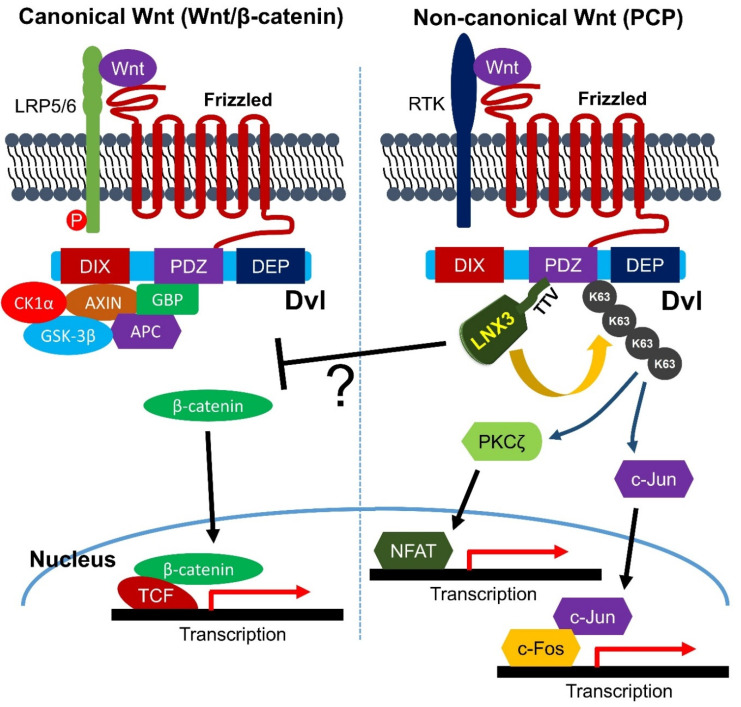
Schematic drawing of differential regulation of canonical and non-canonical Wnt signaling by LNX3. By canonical Wnt stimuli via LRP5/6 and Frizzled receptors, the Dvl recruits the β-catenin destruction complex to the plasma membrane including CK1α and GSK-3β kinases responsible for the β-catenin phosphorylation. As a consequence, the stabilized non-phospho (active) β-catenin is translocated to the nucleus where it binds to and stimulates TCF transcription factors. In noncanonical Wnt signaling, when Wnt ligand binds to Frizzled and RTK receptors, Dvl together with Frizzled undergoes endocytosis to trigger a signaling cascade that drives PKC and c-Jun activation. While PKC activates NFAT transcription factor, c-Jun in conjunction with c-Fos forms AP-1 transcription complex. The C-terminal PDZ-binding motif of LNX3 ended with TTV may directly bind to PDZ domain of Dvl, and then induce K63 linked polyubiquitylation into the Dvl, which is a prerequisite event for the subsequent endocytosis of Frizzled/Dvl complex. Though LNX3 may interfere with canonical Wnt signaling, the detailed mechanism has not been clearly validated yet. PCP; planar cell polarity, Wnt; Wingless and Int-1, LRP5/6; Low density lipoprotein receptor-related protein 5 and 6, Dvl; Dishevelled, RTK; receptor tyrosine kinase, DIX; Dishevelled homologous domain, PDZ; PSD95/DLGA/Zo-1, DEP; Dishevelled/Egl-10/Pleckstrin, GSK-3; Glycogen Synthase Kinase-3, GBP; GSK-3 binding protein, CK1; Casein kinase 1, APC; Adenomatous Polyposis Coli, TCF; T-cell factor, PKC; Protein kinase C, NFAT; Nuclear factor of activated T-cells.

**Table 1 molecules-25-05938-t001:** An explanation of acronyms in alphabetical order.

Acronym	Explanation
BBB	Blood-brain barrier
bHLH	Basic helix-loop-helix
BMP	Bone morphogenetic protein
CA3	Cornu ammonis 3
CAR	Coxsackievirus and adenovirus receptor
Cdx4	Caudal related homeobox transcription factor 4
CLDN16	Claudin-16
CNS	Central nervous system
CNTN3	Contactin-3
DFC	Dorsal forerunner cell
Dvl	Disheveled
E6AP	E6-associated protein
EphB	Ephrin type-B receptor
GlyT2	Sodium- and chloride-dependent glycine transporter 2
Gro/TLE	Groucho/ transducin-like enhancer of split
GST	Glutathione S-transferase
GWAS	Genome-wide association study
HCC	hepatocellular carcinoma
Hdac1	Histone deacetylase 1
HECT	Homologous to the E6-AP Carboxyl Terminus
HOX	Homeobox
HPV	Human papillomavirus
Id2	Inhibitor of DNA binding 2
IL	Interleukin
JAM4	Junctional adhesion molecule 4
KV	Kupffer’s vesicle
LDOC1	Leucine zipper downregulated in cancer 1
LNX	Ligand of numb protein-X
MAGE	Melanoma-associated antigen
MC	Monocarboxylate
MDM2	Murine double minute 2 homolog
MDMX	Murine double minute X
MF	Mossy fiber
MHC	Myogenin heavy chain
MRF	Myogenic regulatory factor
Mrf4	Myogenic regulatory factor 4
MUPP1	Multiple PDZ domain protein 1
Myf5	Myogenic factor 5
MyoD	Myoblast determination protein 1
NEK6	NIMA-related kinases 6, Serine/threonine-protein kinase NEK6
NMDAR	*N*-methyl-D-aspartate receptor
PCP	Planar cell polarity
PDZ	Post-synaptic density protein-95, Disc large tumour suppressor, Zonula occludens-1 (PSD95, DLGA, ZO-1)
PDZK1	PDZ-containing kidney protein 1 (Na(+)/H(+) exchange regulatory cofactor NHE-RF3)
PDZRN	PDZ and RING
pJAK2	Phospho-Janus kinase 2
PKC	Protein kinase C
PPAR	Peroxisome proliferator-activated receptors
RAF-1	V-Raf-1 Murine Leukemia Viral Oncogene Homolog 1
RING	Really interesting new gene
SEMCAP	Semaphoring cytoplasmic domain-associated protein
SINA	Seven in absentia
SMCT1 (SLC5A8)	Sodium-coupled monocarboxylate transporter 1
SNP	Single nucleotide polymorphism
STAT	Signal Transducer and Transcription
STXBP5	Syntaxin binding protein 5
TCF3	T-cell factor 3
TRAF	Tumor necrosis factor receptor (TNF-R)-associated factor
UPS	Ubiquitin proteasome system
Wnt	Wingless and Int-1
Zn	Zinc finger

**Table 2 molecules-25-05938-t002:** Newly discovered LNX1 and LNX2 associated proteins in the past three years (2018–2020).

Associated Proteins	LNX	Domain Involved	Methods Used	Major Functions	Consequences	References
NEK6	LNX1	RING and 3rd PDZ	Co-IP, GST pull down, Yeast two hybrid,	Serine/threonine kinase	K48-linked polyubiquitylated NEK6 undergoes UPS-dependent degradation	[31] (2020)
LDOC 1	LNX1		Co-IP	Tumor suppressor	LNX1 uses LDOC1 as a scaffold protein to indirectly target phospho-JAK2 destruction	[32] (2019)
MDM2	LNX1		Co-IP	E3 ubiquitin ligase	LNX1 may indirectly interact with MDM2	[37] (2019)
p53	LNX1		Co-IP	Tumor suppressor	LNX1 indirectly mediates p53 destruction	[37] (2019)
Connexin 36	LNX1	2nd PDZ	Co-IP, GST pull down, Ni-NTA pull down	Gap junction protein	Ubiquitylated connexin36 undergoes lysosomal-dependent degradation.	[50] (2018)
Connexin 36	LNX2	2nd PDZ	Co-IP, Ni-NTA pull down	Gap junction protein	Ubiquitylated connexin36 undergoes lysosomal-dependent degradation.	[50] (2018)
EphB1	LNX1 p70	N-terminal region ahead of 1st PDZ domain	Co-IP	Receptor tyrosine kinase	Stabilization	[56] (2018)
EphB2	LNX1 p80			Receptor tyrosine kinase	Degradation	[56] (2018)
EphB2	LNX1 p70	2nd PDZ	Co-IP, GST-pull down	Receptor tyrosine kinase	Stabilization	[56] (2018), [57] (2019)
GluN1	LNX1		Co-IP	Glutamate receptor subunit	LNX1 helps form a NMDAR complex by recruiting GluN1 and GluN2B	[57] (2019)
GluN2B	LNX1	1st PDZ	Co-IP, GST pull down	Glutamate receptor subunit	LNX1 helps form a NMDAR complex by recruiting GluN1 and GluN2B	[57] (2019)
GlyT2	LNX1, LNX2	2nd PDZ	Co-IP	Glycine transporter	Polyubiquitylated and degraded	[58] (2019)

**Table 3 molecules-25-05938-t003:** LNX3/PDZRN3 interacting partners.

Associated Proteins	LNX	Domain Involved	Methods Used	Major Functions	Consequences	References
Dvl3	LNX3	C-terminal PDZ binding motif	Co-IP, Yeast two hybrid	Component of Wnt signaling	K63-linked polyubiquitylated Dvl3 undergoes endocytosis with Frizzled to activate non-canonical Wnt signaling	[101] (2014)
SMCT1(SLC5A8)	LNX3	1st PDZ	Co-IP, Yeast two hybrid	Sodium-coupled monocarboxylate (MC) transporter	The potentiation of SMCT1 transporter activity is inhibited by LNX3	[106] (2019)
CLDN16	LNX3	PDZ domain	Co-IP, GST pull down, Yeast two hybrid	Tight junction protein	Monoubiquitylated CLDN16 become subjected to endocytosis	[110] (2017)
MUPP1, PKCζ, Par3	LNX3		Co-IP	Scaffold protein Serine/threonine kinase Polarity protein	Destruction of MUPP1 by LNX3 through UPS inhibits polarity complex formation composed of PAR3, PKCζ, and MUPP1	[113] (2017)
E6	LNX3		GST pull down, Peptide screen of human PDZome	Oncoprotein	E6 targets LNX3 for UPS-dependent destruction	[134] (2015), [152] (2012), [153] (2013)

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
