# Peer review of "The Molecular and Pathophysiological Functions of Members of the LNX/PDZRN E3 Ubiquitin Ligase Family"

_molecules, 2020, doi:10.3390/molecules25245938_

Round 1
Reviewer 1 Report
In this review, the authors introduced structural significance of members of the LNX/PDZRN E3 ubiquitin ligase family firstly. Then they summarized recent and relevant findings on the molecular and cellular functions of the members of the LNX family. Finally, the authors discussed the role of the erroneous regulation of these proteins in disease progression. To my impression, the review is presented in a well-organized and logical manner. All the experimental results show reasonable consistency. In addition, these studies provide insightful knowledge of ligand of Numb protein-X family and will contribute to further studies on its applications. I would therefore strongly recommend this review for publication in Molecules.
Author Response
We are delighted if the reviewer felt fully content to our manuscript. We would be happy if our further rectified work following other reviewer’s valuable comments were more comprehensive for the future audiences. Again we appreciate the praise remark on our manuscript.
Reviewer 2 Report
My comments and manuscript edits are contained in the manuscript pdf.
Since the RING Zn-fingers were referenced, it would be informative to include references that studied the effects of these LNX genes under conditions of zinc deprivation and supplementation.

Author Response
We really appreciate the reviewer of all the rectification of grammar errors and typos of our manuscript. Following the valuable comments of reviewer, we edited several sentences and added a couple of more references related to the putative impact of Zn ion on the function of an in vitro reconstituted E3 ubiquitin ligase.
Reviewer 3 Report
This review by Hong, Won and Ro is focused on biological roles of members of the LNX/PDZRN (ligand of Numb protein X) family. Four members of this family (i.e. LNX1, LNX2, LNX3 and LNX4) are classified as RING E3 ubiquitin ligases due to the presence of a RING domain at their N-termini involved in recruiting an E2 enzyme to catalyze the transfer of ubiquitin on to a substrate protein. Their review highlights recent advances in over the past decade on LNX function as well as a brief overview of LNX domain architecture. While there are many strengths within this review, particularly their discussion on cellular roles of LNX3 and LNX4, most of the review is a rehash of similar discussions in a recent review from 2018 focused on LNX1 and LNX2 functions. This review is also filled with dense text making it difficult to follow and there are no major take-home messages for the reader in its current state. To improve readability, this review would benefit from the inclusion of more figures and/or tables, particularly for a few pathways regulated by LNX3 and LNX4. Smoother segues between the different sections and within each module would also improve the flow of this review. An interaction table to summarize experimentally observed interactions between members of the LNX family with other proteins and/or substrates would also improve this paper. Since each member of the LNX family has multiple PDZ domains, a deeper more focused discussion on these domains is also warranted. The conclusions section should also be reorganized to focus on what’s known, what needs to be addressed, and where the LNX field should focus moving forward. Addressing these issues will help to strengthen this review and make it more accessible to a broader audience.
Concerns:
Many domain acronyms are used throughout this review (ex. SINA). To improve clarity, it is recommended that all of these acronyms be introduced clearly as well as an abbreviation table on the 1st page.
There are a few passages interwoven in the review eluding to the authors unpublished data. While these results are possibly very interesting, they are not currently peer reviewed and should be omitted from the present work.
Molecular structures of LNX1 and LNX2 as well as sequence alignments of predicted domains in LNX3 and LNX4 would strengthen this paper and point to similarities and potential differences. The structure of a PDZ domain binding to a NPAY/NPAF would also be an excellent addition.
Splicing variants - does the loss of the N-terminal RING in any of the LNX family members affect cell homeostasis? Is LNX function lost?
Minor typos:
Terms in latin should be in italics (i.e. via, in vitro, etc.)
Lines 57-58 — needs a smoother segue between UPS and LNX family
Line 123 - not all RING-type E3s form homo- or heterodimers. Some do, others act as monomers.
Line 193 - JAM4 was to prevent in cells
Line 280 - in 2019 [56] further
Line 363 - method may not be inaccurate in cases
Line 765 - evidences that at least a few common substrates
Line 767 - LNX1 is primarily localized
Author Response
We really appreciate all the valuable comments of the reviewer, which would be very helpful to fortify our humble manuscript. We rectified our manuscript following reviewer’s advice as much as possible.
Instead of simply repeating the contents of a previously published review paper (Yong, 2018) regarding to the LNX1 and LNX2, we emphasized novel findings about them in the last three years (2018 ~ 2020), and then moved on to the description of LNX3/4 which has been never summarized in elsewhere. We emphasized the point in the introduction part.
We summarized the main acronyms and their explanation in table 1.
The recently identified binding partners of LNX1 and LNX2 for the past three years were organized in table 2 together with brief descriptions. We also summarized so far reported substrates of LNX3 and LNX4 in table 3.
In order to improve the readability of our manuscript, we added three more tables and two more figures. We also amended several sentences to be more intelligible to the audiences.
We supported a graphical figure (Figure 3) to emphasize the pivotal cellular function of LNX3 especially in Wnt/PCP signaling.
We amended ‘the conclusion part’ following the valuable reviewer’s comments
Following reviewer’s advice, we decided not to mention the unpublished data.
The comparison of protein similarity between individual members of LNX family was shown in Figure 2 with proper description.
We rectified all the grammar errors and typos criticized by the reviewer.
All terms of Latin were changed in italics.
We added description about the various splicing variants of LNX for the future study at the end of conclusion part.
Round 2
Reviewer 3 Report
The revised manuscript is significantly improved. The new figures, tables and revised text improve with the readability and flow of the review. There are still a few minor edits to spelling and grammar that will need to be addressed. The authors should be commended for making these changes as they have made an excellent effort to strengthen their review.
Author Response
We appreciate the additional valuable comments and encouraging remarks of the reviewer.
Following the advices of the reviewer, we rectified some errors in Conclusion part: Figure 2... --> Figure 2b, Figure 2... --> Figure 2c
We also changed the contents of table 2 and table 3 in the order of reference numbers.
Since we do not add any supplementary information, we deleted the remark, 'Supplementary Materials'.
We revised our manuscript with the assistance of a professional English editor to rectify grammar errors (please check the attached 'Certificate').
